# Whole-Body Magnetic Resonance Imaging: Current Role in Patients with Lymphoma

**DOI:** 10.3390/diagnostics11061007

**Published:** 2021-05-31

**Authors:** Domenico Albano, Giuseppe Micci, Caterina Patti, Federico Midiri, Silvia Albano, Giuseppe Lo Re, Emanuele Grassedonio, Ludovico La Grutta, Roberto Lagalla, Massimo Galia

**Affiliations:** 1Sezione di Scienze Radiologiche, Dipartimento di Biomedicina, Neuroscienze e Diagnostica Avanzata, Università degli Studi di Palermo, 90133 Palermo, Italy; albanodomenico@me.com (D.A.); giuseppemicci@gmail.com (G.M.); giuseppe.lore12@gmail.com (G.L.R.); egrassedonio@gmail.com (E.G.); roberto.lagalla@unipa.it (R.L.); massimo.galia@unipa.it (M.G.); 2IRCCS Istituto Ortopedico Galeazzi, Unità Operativa di Radiologia Diagnostica ed Interventistica, 20161 Milan, Italy; 3Ematologia I, Azienda Ospedaliera Ospedali Riuniti Villa Sofia-Cervello, 90146 Palermo, Italy; k.patti@villasofia.it; 4Università degli Studi di Palermo, 90133 Palermo, Italy; silvia.albano@me.com; 5Department of Health Promotion Sciences Maternal and Infantile Care, Internal Medicine and Medical Specialities (ProMISE), University of Palermo, A.O.U.P. Giaccone, 90127 Palermo, Italy; ludovico.lagrutta@unipa.it

**Keywords:** lymphoma, staging, magnetic resonance imaging, diffusion-weighted imaging, whole-body imaging

## Abstract

Imaging of lymphoma is based on the use of ^18^F-fluorodeoxyglucose positron emission tomography/computed tomography (^18^F-FDG-PET/CT) and/or contrast-enhanced CT, but concerns have been raised regarding radiation exposure related to imaging scans in patients with cancer, and its association with increased risk of secondary tumors in patients with lymphoma has been established. To date, lymphoproliferative disorders are among the most common indications to perform whole-body magnetic resonance imaging (MRI). Whole-body MRI is superior to contrast-enhanced CT for staging the disease, also being less dependent on histology if compared to ^18^F-FDG-PET/CT. As well, it does not require exposure to ionizing radiation and could be used for the surveillance of lymphoma. The current role of whole-body MRI in the diagnostic workup in lymphoma is examined in the present review along with the diagnostic performance in staging, response assessment and surveillance of different lymphoma subtypes.

## 1. Introduction

Lymphomas are very common malignant tumors, affecting children, young and old adults, and account for 5–6% of all malignancies [1]. Indeed, Hodgkin lymphoma (HL) and non-Hodgkin’s lymphoma (NHL) are the third most common malignant tumor in children and HL is among the most frequent cancers in pregnancy [2].

Lymphoproliferative disorders consist of different histological subtypes with different prognosis and specific clinical behaviors, which can be divided into three groups, specifically: HL, aggressive lymphomas (a-NHL) and indolent lymphomas (i-NHL) [3]. HL includes two main subgroups: (i) classical HL that represents over 90% of HL patients, high ^18^F-fluorodeoxyglucose (^18^F-FDG) uptake at ^18^F-FDG positron emission tomography/computed tomography (^18^F-FDG-PET/CT) and aggressive clinical picture; and (ii) nodular lymphocyte-predominant HL that generally shows indolent behavior and low, if any, FDG uptake at ^18^F-FDG-PET/CT [4].

The a-NHL subtypes are characterized by fast tumor growth and need of immediate treatment. Among them, diffuse large B-cell lymphoma (DLBCL), mantle cell lymphoma, Burkitt lymphoma and peripheral T-cell lymphomas represent the main aggressive subtypes, with DLBCL being the most common and accounting for 30% of all NHLs [5]. DLBCL displays high FDG uptake similar to most aggressive lymphomas [6]. 

The i-NHL subgroup consist of slow-growing malignancies characterized by prolonged natural history and generally few, if any, clinical symptoms [7]. Follicular lymphoma is the most common i-NHL and accounts for about 20% of all NHLs. In the majority of i-NHLs, immediate treatment is not needed, and watchful waiting is considered an optimal option in most cases [8]. 

Imaging of lymphoma is based on the use of ^18^F-FDG-PET/CT and/or contrast enhanced CT for FDG-avid lymphomas, whereas contrast enhanced CT is recommended for non-FDG-avid subtypes [9,10]. Concerns have been raised regarding radiation exposure related to multiple imaging scans performed for staging and follow-up in patients with cancer; in fact, the association with an increased risk of secondary neoplasms in patients with lymphoma has been established [11,12]. This has spurred increased interest in the use of whole-body magnetic resonance imaging (whole-body MRI) as a radiation-free alternative to standard imaging examinations to evaluate patients with lymphoma [12,13]. To date, lymphoproliferative disorders are among the most common indicators in performing whole-body MRI [13,14]. The current role of whole-body MRI in the diagnostic workup in lymphoma is examined in the present review along with the diagnostic performance in staging, response assessment and surveillance of different lymphoma subtypes.

## 2. Current Recommendations for Imaging of Lymphoma

To date, whole-body MRI is not considered in the international guidelines for staging or response assessment in patients with lymphoma [9]. 

The ^18^F-FDG-PET/CT is the preferred imaging modality for the staging of patients with HL and FDG-avid NHL subtypes [9]. In these patients, ^18^F-FDG-PET/CT is usually associated with a contrast enhanced CT to obtain an accurate measurement of the nodal localizations, to characterize focal parenchymal lesions, to distinguish between viscera and lymphadenopathy, and to evaluate thrombosis or great thoracic vessels compression. Furthermore, contrast enhanced CT is usually performed as a pre-operative planning examination in patients who are eligible for a radiotherapy treatment. 

Contrast enhanced CT remains the recommended imaging modality for staging patients with non–FDG-avid subtypes or with lymphomas with variable FDG avidity (namely chronic lymphatic leukemia/small lymphocyte lymphoma, lymphoplasmacytic lymphoma/Waldestrom macroglobulinemia, mycosis fungoides and marginal zone lymphoma). 

Concerning the assessment of the bone marrow involvement, bone marrow biopsy has been the gold standard technique in the lymphoma staging for a long time [15]. However, bone marrow biopsy is often done even if the likelihood of bone marrow involvement is quite low. Further, the high sensitivity of ^18^F-FDG-PET/CT in detecting such involvement has raised the issue of whether to continue, or not, to routinely perform bone marrow biopsy in patients with FDG-avid subtypes [16]. The current indication for these subtypes is to not perform bone marrow biopsy in patients with HL and to perform it in those patients with DLBCL and negative ^18^F-FDG-PET/CT with the purpose to search for low-volume diffuse bone marrow involvement that can be missed in some cases [9]. 

^18^F-FDG-PET/CT is the gold standard in the evaluation of the response of FDG-avid subtypes to therapies. It is performed as an interim evaluation during chemotherapy and at the end of the treatment, by using a semiquantitative five points scale to assess and quantify the presence of active disease. The absence of uptake in the FDG-avid subtypes is considered a complete response even with residual masses. Instead, the response assessment in the non–FDG-avid subtypes or with variable avidity is carried out with contrast enhanced CT. However, in this case, a reduction in mass with residual tissue can be considered at most a partial response in absence of histological confirmation [9]. 

Regarding the follow-up of lymphoma patients, the role of imaging is controversial. Several studies have failed in demonstrating an advantage of using ^18^F-FDG-PET/CT after treatment in the surveillance of patients with lymphoma [17]. The false positive rate has been higher than 20%, increasing the number of unnecessary exams, the risks related to exposure to ionizing radiation and the burdensome anxiety of patients. Indeed, contrast enhanced CT is generally recommended every 3–6 months in the first two years after the end of the treatment, as well as according to the lymphoma subtype and the pre-therapy prognostic factors, and with larger time intervals [9].

## 3. Whole-Body MRI: General and Technical Aspects

Whole-body MRI scan is based on moving-table acquisitions, multi-channel surface coils and parallel imaging acquisition, which help achieve high spatial resolution with good signal-to-noise ratio throughout the body. Technological developments have reduced the scanning time for this examination, such that diffusion-weighted imaging (DWI) can be performed along with morphologic sequences in a reasonably short acquisition time (30–45 min), making this exam also feasible for older patients with suboptimal performance status [18]. The anatomical coverage of a whole-body MRI examination is usually from the skull base to mid-thigh, as that of PET/CT, but also including the upper limbs; although a true “whole-body” scan from head to toe can be performed [19].

To date, no clear guidelines have been published on whole-body MRI protocols to be used in patients with lymphoma given that there is a lack of agreement on this point. Different approaches have been proposed in previous studies, in terms of type of sequences, acquisition planes and contrast media injection. In most studies, imaging protocols include unenhanced turbo spin-echo T1-weighted and T2-weighted images, short tau inversion recovery (STIR) and DWI sequences [18,20]. In this regard, different b-values have been used on previous papers to acquire DWI sequences, taking advantage from the high cellularity and nuclear-to-cytoplasm ratio of locations of lymphoma that lead to restricted patterns of diffusion, thereby producing high signal intensity on DWI image and low apparent diffusion coefficient (ADC) values [18]. DWI is routinely used in cancer patients to assess tumor cellularity and to improve tumor detection, particularly to identify metastatic locations of disease [21,22]. In patients with lymphoma, whole-body DWI is generally performed using sections of 5 to 7 mm with axial orientation, during free breathing, using a single-shot spin-echo planar imaging acquisition. Fat suppression with STIR is strongly recommended to obtain homogeneous fat signal saturation in DWI sequences using such large fields of view. Two b-values are usually sufficient to calculate ADC, although more b-values mean higher reliability of quantitative assessment of diffusion restriction. The lowest b-value is usually in the range from 50 to 100 s/mm^2^ to minimize perfusion related signals, while the highest is typically between 800 and 1000 s/mm^2^, allowing good detection of hyper-cellular lesions with good signal-to-noise ratio. Dixon sequences are based on chemical shifts and are more used to quickly acquire T1-weighted images due to their capability of deriving multiple images, namely: in phase, out of phase, fat only and water only, in a single acquisition providing homogeneous water/fat suppression with great advantages for bone marrow assessment [21].

Different whole-body MRI criteria have been postulated to identify lymph nodal involvement, in addition to standard size criteria (longest diameter > 1.5 cm) [20]. Lymph nodes can be considered involved when: (i) the DWI signal is higher than that of the spinal cord; (ii) the DWI signal remains high at higher b-values, with restriction confirmed by low ADC or in the presence of central necrosis, regardless the nodal size; and (iii) when lymph nodes coalesce into large nodal masses [14]. Concerning quantitative evaluation of DWI, despite ADC measurements on lymph nodes having been shown to be reproducible, no standardized cut-off values for ADC currently exist to differentiate normal lymph nodes from locations of lymphoma. Further, it has not been defined whether average or minimum ADC values should be used for this purpose.

Despite its excellent diagnostic performance being proven by several studies, whole-body MRI has shown various weaknesses in the assessment of small thoracic lesions (hilar, mediastinal and pulmonary) and those detected in tissues with physiologically limited anisotropic diffusion patterns (such as spleen, nervous system and renal parenchyma) [23,24]. The former limitation is due to artifacts observed on DWI sequences related to cardiac pulsation and breathing that may alter the calculation of ADC, while the latter is related to challenging detection of lymphoma locations in organs with low ADC values in normal conditions [23]. In these settings, the acquisition of 3D T1-weighted images after the administration of gadolinium based contrast agents during whole-body MRI may improve the accuracy of identifying parenchymal lesions [25]. However, some authors believe that unenhanced morphologic evaluation with standard sequences associated with DWI is sufficiently effective to obtain a comprehensive and reliable assessment of lymphoma staging [26]. Moreover, this view is further supported by the growing evidence of gadolinium accumulation in human tissues, whose clinical implication still needs to be clarified [27,28]. Regarding the potential artifacts encountered on whole-body MRI scans, the current application of 3T units should be mentioned. Indeed, despite the potential of 3T scanners in providing a higher signal-to-noise ratio and improving tumor detection, whole-body MRI at 3T is partly affected by image shearing, geometric distortion, chemical shifts and ghosting artifacts, particularly affecting image reformats. Azzedine et al. compared 1.5T and 3T whole-body MRI for staging 23 patients with lymphoma in a prospective study, reporting very high and similar diagnostic performance, except for major artifacts in 2 out of 23 patients at 3T [29]. To date, no studies have proven the superiority of 3T whole-body MRI over 1.5T.

Table 1 summarizes strengths and weak points of the main imaging modalities used in patients with lymphoma.

## 4. Staging of Lymphoma

To date, ^18^F-FDG-PET/CT represents the reference standard for staging and response assessment in FDG-avid lymphomas. Contrast enhanced CT is preferred to study variably/low FDG-avid lymphomas, in which, however, ^18^F-FDG-PET/CT is often performed along with contrast enhanced CT [9].

Concerning the role of whole-body MRI in lymphoma staging, good to excellent agreement between whole-body MRI and ^18^F-FDG-PET/CT has been reported in the literature for the detection of both nodal and extra-nodal locations of disease [30]. Whole-body MRI has been shown to be lightly inferior to ^18^F-FDG-PET/CT for staging FDG-avid subtypes; nevertheless, several authors have demonstrated that whole-body MRI might be superior to both ^18^F-FDG-PET/CT and contrast enhanced CT in lymphomas with low or no FDG avidity [31,32]. 

Previous papers have proven high accuracy of whole-body MRI for disease staging in patients with HL [24,33]. In a prospective trial of 140 patients with lymphoma, Mayerhoefer and colleagues reported very high agreement (kappa = 0.92 for lymphomas, which presented high FDG avidity; kappa = 0.89 for lymphomas with variable FDG avidity) between whole-body MRI and ^18^F-FDG-PET/CT in staging patients with HL [24]. Albano and coworkers assessed the diagnostic performance of whole-body MRI using ^18^F-FDG-PET/CT as a reference standard to stage 68 patients with FDG-avid lymphoma (37 with HL) [29]. The authors reported excellent agreement between the two imaging modalities for HL staging (kappa = 0.92) similarly to data reported by Mayerhoefer and colleagues. In the series by Albano and coworkers, whole-body MRI over-staged a periaortic lymph node in one patient with HL, while whole-body MRI under-staged a lung lesion interpreted as a hilar lymph node in another patient with HL [33]. Figure 1 show a case of whole-body MRI used for staging patients with HL.

Additionally, Lin and colleagues proved the value of whole-body MRI for the staging of patients with DLBCL, reporting agreement between whole-body MRI and ^18^F-FDG-PET/CT in the staging 93% of patients [34]. Even Abdulqadhr and coworkers reported excellent results of whole-body MRI for the staging of aggressive lymphomas, showing complete agreement in staging 18 patients with aggressive subtypes (DLBCL, primary mediastinal B-cell lymphomas, anaplastic large cell lymphoma and T-cell lymphoma) [35].

Kwee et al. also reported excellent results comparing whole-body MRI and contrast enhanced CT in 104 patients with lymphoma. The authors emphasized that whole-body MRI improved the diagnosis in staging 30% of patients with aggressive lymphomas detecting hidden bone marrow involvement, but it missed pleural, lung and lymph nodal involvements in about 7% of cases [36]. Figure 2 shows a case of aggressive lymphoma (peripheral T-cell lymphoma) staged with whole-body MRI.

An interesting point that has emerged over the last years is the application of whole-body MRI in the staging of i-NHL, given that previous papers reported that whole-body MRI is even superior to contrast enhanced CT [35,37]. Balbo-Mussetto et al. and Abdulqadhr et al. showed that whole-body MRI was superior to contrast enhanced CT and ^18^F-FDG-PET/CT, respectively [35,37] reaching a correct up-staging of disease in patients with i-NHL. Furthermore, Mayerhoefer and colleagues showed the higher diagnostic performance of whole-body MRI in patients with mucosa-associated lymphoid tissue lymphomas, which are characterized by variable FDG avidity in ^18^F-FDG-PET/CT, reporting 94.4% sensitivity of whole-body MRI, 60.9% sensitivity of ^18^F-FDG-PET/CT and 70.7% sensitivity of contrast enhanced CT (70.7%) [23]. Stecco and colleagues focused their study on patients with gastrointestinal lymphomas; the authors reported excellent results comparing whole-body MRI and ^18^F-FDG-PET/CT, and including 12 i-NHLs (kappa = 0.87) [38].

Another crucial field of interest is the non-invasive assessment of bone marrow involvement through imaging examinations [39]. Several papers have demonstrated the strong potential of whole-body MRI to identify focal and diffuse bone marrow involvement in patients with lymphoma, especially in comparison with contrast enhanced CT, which have well established limits in the evaluation of bone locations of lymphoma that can present with permeative patterns or slightly sclerotic/lytic lesions [32,40]. This might be particularly important in those patients with lymphoma in which focal involvement can be missed by blinded unilateral bone marrow biopsy of the iliac crest [41]. Interestingly, higher risk of disease progression and death in patients has been reported with DLBCL with negative bone marrow biopsy and positive whole-body MRI for marrow involvement if compared with negative whole-body MRI [42]. However, it should be considered that it seems that both whole-body MRI and ^18^F-FDG-PET/CT have lower accuracy in the detection of bone marrow involvement in patients with i-NHL [40].

## 5. Response to Therapy, Surveillance and Follow-Up

^18^F-FDG-PET/CT is the standard imaging modality for post-treatment evaluation of HL and aggressive lymphomas, and it is often applied for interim assessment during chemotherapy to predict disease response early [9,42,43].

Albano and colleagues found that interim DWI after a few courses of chemotherapy might be used to identify those HL locations responsive to systemic treatment [44], reporting a significant increase of ADC in responding lesions, similar to the results reported by Horger and coworkers in HL and NHL patients [45]. Notably, Latifoltojar et al. highlighted that whole-body MRI might underestimate the response of extra-nodal locations of disease after treatment [46]. The increase of ADC values in enlarged masses after treatment has also been reported in patients with DLBCL, highlighting the poor value of the mere size evaluation of locations of lymphoma after therapy [47,48]. In this regard, De Paepe and colleagues investigated 14 patients with aggressive lymphoma with whole-body MRI before, after two courses and at the end of treatment. The authors reported significantly different ADC values from responder to non-responder lesions, with DWI having shown a 100% negative predictive value and correlation with progression free survival (*p* < 0.05) [48]. On the other hand, the morphologic evaluation of lymphomatous lesions to assess size changes was not able to assess the early response to therapy [43]. Furthermore, Mayerhoefer and colleagues recently demonstrated the similar diagnostic performance of whole-body MRI and ^18^F-FDG-PET/CT to assess the response during and after chemotherapy in 64 patients with lymphoma [49].

Young patients with lymphoma have prolonged overall survival (90–95% at 10 years) [50], but ^18^F-FDG-PET/CT and contrast enhanced CT are still recommended to follow these patients after treatment [51]. The concern about the high radiation exposure of young patients with long life expectancy has encouraged some authors to support the use of whole- body MRI to monitor lymphomas in watchful waiting or in complete remission [52,53]. Nevertheless, no previous studies have compared the clinical impact of imaging surveillance performed with whole-body MRI and ^18^F-FDG-PET/CT or contrast enhanced CT on patients in watchful waiting or in complete remission.

Further, whole-body MRI is also highly effective to identify osteonecrotic lesions occurring after chemotherapy, including high doses of corticosteroids [54,55]. Indeed, whole-body MRI enables the assessment of the whole skeletal system and is the best imaging modality to early detect multifocal osteonecrosis, given that osteonecrosis can be easily distinguished from locations of lymphoma on MRI. Further, an association between steroid dose, number of chemotherapy cycles and risk of osteonecrosis has been demonstrated in patients with lymphoma, strengthening the potential of this imaging tool for the surveillance of patients with lymphoma [54]. Figure 3, Figure 4 and Figure 5 show examples of whole-body MRI used for pre- and post-treatment staging of patients with NHL (Mantle Cell Lymphoma, Follicular Lymphoma and DLBCL).

## 6. Future Perspectives

Among the potential perspectives of the use of whole-body MRI to image patients with lymphoma, new and interesting fields of application include the computer assisted diagnosis, the texture analysis and the radiomics [55,56,57,58,59]. Several authors proposed the use of CAD to improve the lesion (damage) segmentation and the ADC value calculation in other tumors [55]. Colombo and colleagues obtained a good intra- and inter-observer reproducibility in the semi-automated segmentation of the ADC values of bone metastases in whole-body MRI performed for the staging of breast and prostate cancers [55].

Brancato et al. tested an automatic approach based on the whole-body DWI sequences for the prediction and assessment of response to treatment in 20 patients with HL [56]. The authors automatically extracted values of volume diffusion and its associated histogram features by whole-body DWI images and evaluated their utility in predicting and assessing interim and end of treatment response. In their pilot study, Spijkers et al. merged in post-processing high b value DWI sequences and T2 weighted images into a color parametric map, showing that the addition of fused images to whole-body MRI protocols for staging of pediatric HL might be of potential additional value [57].

Regarding the application of artificial intelligence on whole-body MRI performed in patients with lymphoma, only a few studies have investigated this opportunity. In particular, radiomics has been recently used in several other tumors to extract a huge amount of image features, which can only be identified by using computers. Texture analysis can be used to assess the spatial pattern and arrangement of pixel intensities in images to detect thetumoral heterogeneity, which several authors have shown, to have non-negligible correlations with tumor behavior, prognosis and response to treatment. In this setting, a relatively recent work by De Paepe et al. has shown that the first order texture analysis of the ADC values improves the diagnostic performance of DWI in the characterization of locations of lymphoma if compared to the simple ADC average value calculation [58]. Similarly, a recent interesting paper on the texture analysis of post contrast T1 weighted images reported the high diagnostic performance in differentiating follicular lymphoma from DLBCL that can be a valuable tool to identify the aggressive transformation of i-NHL [59].

## 7. Conclusions

Whole-body MRI is an imaging modality with impressive potential in evaluating patients with lymphoma, although its role in the diagnostic workup has not been clearly defined. Whole-body MRI is superior to contrast enhanced CT in the staging of the disease, and also in being less dependent on histology if compared to ^18^F-FDG-PET/CT. Moreover, it does not require exposure to ionizing radiation and could be used for the surveillance of patients with lymphoma in complete remission and in watchful waiting. Future studies will clarify the potential of interesting new perspectives that include the aid of post-processing to diagnosis and the application texture analysis on whole-body MRI. In conclusion, the use of whole-body MRI should be further explored and viewed with particular interest by the scientific community to establish how to add whole-body MRI in the diagnostic path of patients with lymphoma.

## Figures and Tables

**Figure 1 diagnostics-11-01007-f001:**
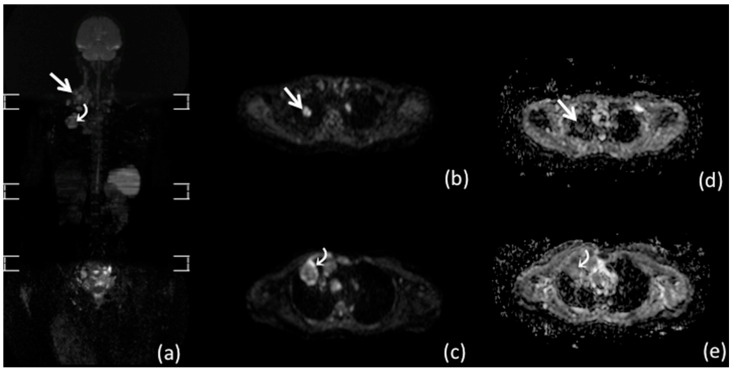
Whole-body MRI of a 28-year-old man with Hodgkin Lymphoma. Coronal maximum intensity projection (MIP) b = 800 diffusion-weighted imaging (DWI) (**a**), axial b 800 DWI images of the chest (**b**,**c**), and axial ADC maps of the chest (**d**,**e**), show a lung location in the apical segment of the upper lobe of the right lung (curved arrow in (**a**); arrow in (**b**,**d**)) and multiple nodal locations at internal mammary level (arrow in (**a**); curved arrow in (**c**,**e**)).

**Figure 2 diagnostics-11-01007-f002:**
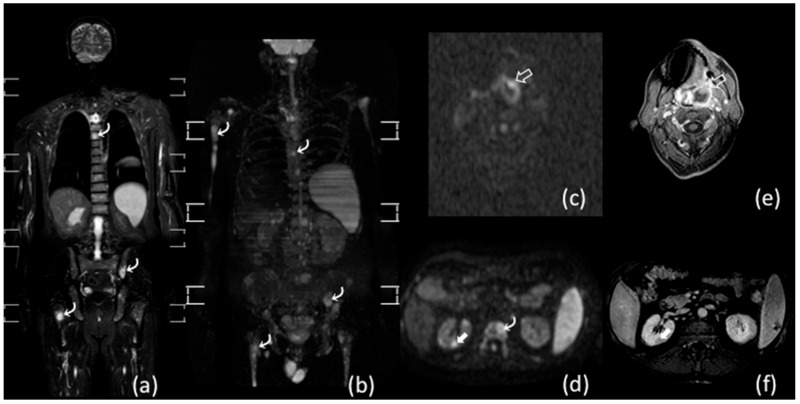
Whole-body MRI of a 65-year-old man with peripheral T-cell lymphoma. Coronal short tau inversion recovery (STIR) (**a**), maximum intensity projection (MIP) b 800 diffusion-weighted imaging (DWI) (**b**), axial b 800 DWI (**c**,**d**), and axial 3D GRE T1 weighted images after intravenous contrast media injection (**e**,**f**) show multiple extranodal locations in the left palatine tonsil (void arrows in (**c**,**e**)), in the right kidney (white arrows in (**d**,**f**)) and in the bone marrow (curved arrows in (**a**,**b**,**d**)).

**Figure 3 diagnostics-11-01007-f003:**
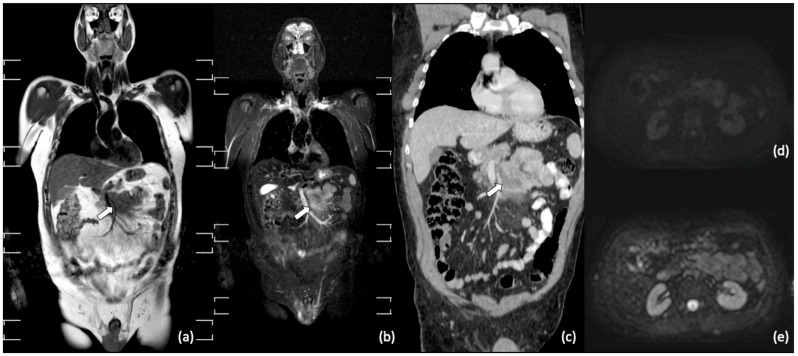
Whole-body MRI and contrast enhanced CT after chemotherapy treatment of a 46-year-old man with Follicular Lymphoma. Coronal T1-weighted (**a**); coronal short tau inversion recovery (STIR) (**b**); coronal MPR CT portal phase(**c**); axial b800 DWIBS (**d**); axial b50 DWIBS (**e**). Note residual mass in mesenteric site (arrows) without signal restriction in DWI (**d**,**e**).

**Figure 4 diagnostics-11-01007-f004:**
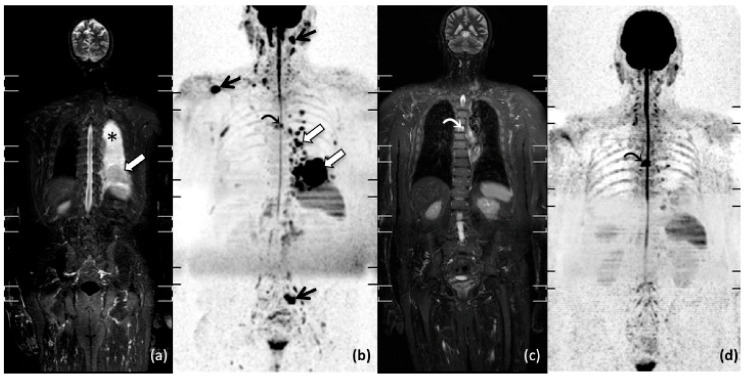
Whole-body MRI of a 62-year-old man with pleural non-Hodgkin Mantle Cell Lymphoma. Pre-treatment coronal short tau inversion recovery (STIR) (**a**), maximum intensity projection (MIP) b 800 grey-scale inverted DWI (**b**); post-treatment coronal STIR (**c**); and MIP b 800 grey-scale inverted DWI (**d**). Pre-treatment images show multiple pleural (white arrow in (**a**,**b**)) and nodal (black arrow in (**b**)) locations of disease with complete response after treatment (**c**,**d**); also note the presence of pleural effusion (* in (**a**)), and a vertebral hemangioma with no changes after chemotherapy (curved arrow in (**b**–**d**)).

**Figure 5 diagnostics-11-01007-f005:**
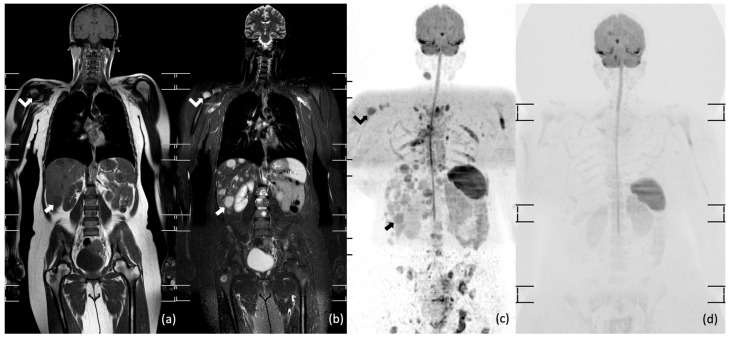
Whole-body MRI of a 36-year-old woman with non-Hodgkin diffuse B-cell lymphoma, treated with six courses of DA-REPOCH. Pre-treatment coronal T1-weighted (**a**); pre-treatment coronal short tau inversion recovery (STIR) (**b**); pre-treatment maximum intensity projection (MIP) b 800 grey-scale inverted DWI (**c**); post-treatment maximum intensity projection (MIP) b 800 grey-scale inverted DWI (**d**). Pre-treatment images show multiple hepatic (white arrow in (**a**,**b**) and black arrow in (**c**) and bone (white curved arrow in (**a**,**b**), black curved arrow in **c**) locations of disease with complete response after treatment (**d**).

**Table 1 diagnostics-11-01007-t001:** Strengths and weak points of the main imaging modalities used in patients with lymphoma.

Imaging Modality	Strengths	Weak Points
Whole-body MRI	High contrast resolution in soft tissues and bone marrow	MRI contraindications (i.e., pace-maker, claustrophobia)
	Cellularity assessment through diffusion- weighted imaging	Limited availability and less performed by general radiologists
	Neither contrast injection nor radiation exposure	Lower diagnostic performance in lung locations of disease
	It can be performed in pregnant patients	Long acquisition and reporting time
Contrast enhanced CT	Widely available and standard acquisition protocol	No functional or metabolic information
	High spatial resolution	Contrast media administration
	Short acquisition time	Radiation exposure
^18^F-FDG-PET/CT	Metabolic evaluation with recognized SUV_max_ cutoff	Histology dependent, some subtypes do not work for FDG uptake
	Standardized acquisition and reporting (Deauville Score)	High burden of ionizing radiations
	Wide availability	Long acquisition time

## Data Availability

Not applicable.

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
