# Peer review of "Whole-Body Magnetic Resonance Imaging: Current Role in Patients with Lymphoma"

_diagnostics, 2021, doi:10.3390/diagnostics11061007_

Round 1

Reviewer 1 Report

In this review article, the authors summarized the role of whole-body magnetic resonance imaging (MRI) in patients with lymphoma.  I thought the text was reasonably well written and informative.

Comments;

  1. Figures are well informative for the readers to understand the role of MRI, but they are not described in the manuscript. Additionally, the comparison of several images including MRI, 18F-FDG-PET/CT and enhanced CT will help the readers understand utility of MRI, probably in indolent lymphoma subtypes.

  1. Summary tables describing advantages and disadvantages of MRI, 18F-FDG-PET/CT and enhanced CT will be helpful for the readers.

  1. For indolent lymphomas, are there any studies showing that the detection of tumor dissemination by MRI more precisely predict the clinical course?

Author Response

We thank you for your positive reply. 

Reviewer #1:

Comment 1: In this review article, the authors summarized the role of whole-body magnetic resonance imaging (MRI) in patients with lymphoma. I thought the text was reasonably well written and informative.

Answer 1: we thank the reviewer for his positive comment.

Comment 2: Figures are well informative for the readers to understand the role of MRI, but they are not described in the manuscript. Additionally, the comparison of several images including MRI, 18F-FDG-PET/CT and enhanced CT will help the readers understand utility of MRI, probably in indolent lymphoma subtypes.

Answer 2: we have referenced the figures along the text and we have added a figure as suggested.

Comment 3: Summary tables describing advantages and disadvantages of MRI, 18F-FDG-PET/CT and enhanced CT will be helpful for the readers.

Answer 3: We agree with the reviewer’s comment. We have added a table to summarize strength and weak points of these three imaging modalities.

Comment 4: For indolent lymphomas, are there any studies showing that the detection of tumor dissemination by MRI more precisely predict the clinical course?

Answer 4: to the best of our knowledge, the predictive role of MRI in indolent lymphoma has not still proven. Only one study by Wu et al investigated the potential role of texture analysis of post-contrast T1-weithed MR image to identify the aggressive transformation of i-NHL. This paper has been discussed and cited at the end of the paragraph “Future perspectives”.

Reviewer 2 Report

This is an adequate review of whole-body MRI of lymphoma patients, written by authors knowledgeable of the field. I think the manuscript provides a handy overview, which is easy to follow. I do miss statements concerning pros and cons of performing whole-body MRI at 1.5T vs. 3T systems, so I would suggest the authors to add this information to the manuscript, possibly with a recommendation of the preferred field strength with current knowledge.

Specific points:

  1. The manuscript contains four figures, none of which are referenced in the text. Are these images from the authors’ institution? Please add the sources of the images to the figure legends, including a copyright statement where appropriate. Please also make references to the figures in the running text.
  2. The references are not written in a consistent format. Please write out the full author list and include the first letter of the first name. Of 68 references, 11 are self-citations, a rather high number. The authors may consider if all are necessary.
  3. Page 2, Line 66: Please add references to “international guidelines for staging or response assessment in patients with lymphoma.” One such reference may be “Recommendations for initial evaluation, staging, and response assessment of Hodgkin and non-Hodgkin lymphoma: the Lugano classification” by Cheson et al. (PMID 25113753).
  4. P2 L80: “Concerning the assessment…” instead of “For what concerns the assessment…”. Also avoid “For what concerns…” on P3 L145, P4 L176 and P8 L309.
  5. P2 L95: “…a mass in reduction but persistent can be considered…”. This sentence is strange. Does the author mean “a reduction in mass”? Please rephrase the sentence for clarity.
  6. P3 L112: in reference [14], I cannot find a statement of the total scan time for a whole-body examination with both DWI and morphological imaging. Reference [16] seems to be more relevant for this purpose.
  7. P5, L185 and elsewhere: use “kappa” instead of “k” to represent agreement.
  8. P8 L294-298: Please add references already here (first two sentences of the paragraph).

Author Response

We thank you for your positive reply. 

Comment 1: This is an adequate review of whole-body MRI of lymphoma patients, written by authors knowledgeable of the field. I think the manuscript provides a handy overview, which is easy to follow. I do miss statements concerning pros and cons of performing whole-body MRI at 1.5T vs. 3T systems, so I would suggest the authors to add this information to the manuscript, possibly with a recommendation of the preferred field strength with current knowledge.

Answer 1: we thank the reviewer for his positive comment. We have added the information concerning the value of 1.5 and 3T units at the end of the paragraph 3. Whole body MRI: general and technical aspects: “Regarding the potential artifacts encountered on whole body MRI scans, the current application of 3T units should be mentioned. Indeed, despite 3T scanners may provide higher signal-to-noise ration potentially improving tumor detection, whole body MRI at 3T is partly affected by image shearing, geometric distortion, chemical shift and ghosting artifacts, particularly affecting images reformats. Azzedine et al compared 1.5T and 3T whole body MRI for staging 23 patients with lymphoma in a prospective study reporting very high and similar diagnostic performance, but major artifacts in 2 out of 23 patients at 3T [Clin Imaging 2015;39(1):104-9. doi: 10.1016/j.clinimag.2014.06.017]. To date, no studies have proven the superiority of 3T whole body MRI over 1.5T.”

Comment 2: The manuscript contains four figures, none of which are referenced in the text. Are these images from the authors’ institution? Please add the sources of the images to the figure legends, including a copyright statement where appropriate. Please also make references to the figures in the running text.

Answer 2: we have referenced the figures in the text. We did not specify the source of these images given that they are from our Institution.

Comment 3: The references are not written in a consistent format. Please write out the full author list and include the first letter of the first name. Of 68 references, 11 are self-citations, a rather high number. The authors may consider if all are necessary.

Answer 3: we agree with the reviewer’s comment. As suggested, we have edited the references to be consistent and we have deleted some self-citations.

Comment 4: Page 2, Line 66: Please add references to “international guidelines for staging or response assessment in patients with lymphoma.” One such reference may be “Recommendations for initial evaluation, staging, and response assessment of Hodgkin and non-Hodgkin lymphoma: the Lugano classification” by Cheson et al. (PMID 25113753).

Answer 4: As suggested, we have added this reference.

Comment 5: P2 L80: “Concerning the assessment…” instead of “For what concerns the assessment…”. Also avoid “For what concerns…” on P3 L145, P4 L176 and P8 L309.

Answer 5: we thank the reviewer for his suggestions, we have modified the text accordingly.

Comment 6: P2 L95: “…a mass in reduction but persistent can be considered…”. This sentence is strange. Does the author mean “a reduction in mass”? Please rephrase the sentence for clarity.

Answer 6: as suggested, we have rephrased this sentence to better convey our message.

Comment 7: P3 L112: in reference [14], I cannot find a statement of the total scan time for a whole-body examination with both DWI and morphological imaging. Reference [16] seems to be more relevant for this purpose.

Answer 7: we thank the reviewer for his suggestion, we have used the reference 16 in this sentence.

Comment 8: P5, L185 and elsewhere: use “kappa” instead of “k” to represent agreement.

Answer 8: as suggested we have replaced k with kappa.

Comment 9: P8 L294-298: Please add references already here (first two sentences of the paragraph).

Answer 9: we have added the references as suggested.